# The Burden of Potentially Inappropriate Medications in Chronic Polypharmacy

**DOI:** 10.3390/jcm9113728

**Published:** 2020-11-20

**Authors:** Jordan Guillot, Sandy Maumus-Robert, Alexandre Marceron, Pernelle Noize, Antoine Pariente, Julien Bezin

**Affiliations:** 1INSERM U1219, Bordeaux Population Health, Team Pharmacoepidemiology, University of Bordeaux, F-33000 Bordeaux, France; sandy.robert@u-bordeaux.fr (S.M.-R.); pernelle.noize@u-bordeaux.fr (P.N.); antoine.pariente@u-bordeaux.fr (A.P.); julien.bezin@u-bordeaux.fr (J.B.); 2Service de Pharmacologie Médicale, Pôle de Santé Publique, CHU de Bordeaux, 33000 Bordeaux, France; alexandre.marceron1@gmail.com

**Keywords:** polypharmacy, appropriateness, potentially inappropriate medication, drug utilization study, observational study

## Abstract

We aimed to describe the burden represented by potentially inappropriate medications (PIMs) in chronic polypharmacy in France. We conducted a nationwide cross-sectional study using data from the French National Insurance databases. The study period was from 1 January 2016 to 31 December 2016. Chronic drug use was defined as uninterrupted daily use lasting ≥6 months. Chronic polypharmacy was defined as the chronic use of ≥5 medications, and chronic hyperpolypharmacy as the chronic use of ≥10 medications. For individuals aged ≥65 (older adults), PIMs were defined according to the Beers and Laroche lists, and for individuals aged 45–64 years (middle-aged) PIMs were defined according to the PROMPT (Prescribing Optimally in Middle-aged People’s Treatments) list. Among individuals with chronic polypharmacy, 4009 (46.2%) middle-aged and 18,036 (64.8%) older adults had at least one chronic PIM. Among individuals with chronic hyperpolypharmacy, these figures were, respectively, 570 (75.0%) and 2544 (88.7%). The most frequent chronic PIM were proton pump inhibitors (43.4% of older adults with chronic polypharmacy), short-acting benzodiazepines (older adults: 13.7%; middle-aged: 16.1%), hypnotics (6.1%; 7.4%), and long-acting sulfonylureas (3.9%; 12.3%). The burden of chronic PIM appeared to be very high in our study, concerning almost half of middle-aged adults and two-thirds of older adults with chronic polypharmacy. Deprescribing interventions in polypharmacy should primarily target proton pump inhibitors and hypnotics.

## 1. Introduction

Multimorbidity is, nowadays, common and increases with age [1]. Polypharmacy, defined as the simultaneous use of many drugs, is linked to multimorbidity and is thus consistently rising, especially in the elderly [2,3,4]. While less studied, multimorbidities and their accompanying polypharmacy are already highly prevalent in middle-aged individuals [5]. Though potentially motivated by the therapeutic management of coexisting diseases, polypharmacy, which is usually defined as the concomitant use of five drugs or more [4,6], is associated with harmful events such as hospitalizations, falls, and even death [7]. Ensuring that polypharmacy is limited to what is needed for patients’ best care is thus an important challenge. Limiting its inflation could indeed be of important matter in terms of public health, as inappropriate prescribing seems to increase with the number of chronic drugs [8].

Potentially inappropriate medications (PIMs) are defined as drugs for which the risks outweigh the potential benefits, especially when more efficient alternatives are available [9]. The use of PIMs in the population has been extensively studied, in particular in the elderly, where the prevalence of their use is high, as is the risk of related adverse events [10,11,12,13,14,15,16]. Conversely, the literature is scarce concerning its assessment in middle-aged adults, and especially in the context of polypharmacy. Yet, the prevalence of multimorbidity in the middle-aged attracts study in this population [1], especially in that those with polypharmacy should be the more likely to have PIMs [8].

The aim of this study was to evaluate, according to age, the burden of chronic potentially inappropriate medications in adults with chronic polypharmacy in the French population.

## 2. Methods

### 2.1. Study Design

We conducted a nationwide cross-sectional study using data from the French National Insurance databases.

### 2.2. Data Source

The EGB (Echantillon Généraliste de Bénéficiaires) is a nationwide permanent 1/97th random sample of the databases of the French Health Care Insurance System which covers 98.8% of the French population. It is representative of the general population in terms of age, gender, and medical expenses. It contains individual anonymous information on all outpatient reimbursed healthcare expenditures; the registration status for a list of 30 specifically individualized chronic diseases; and hospital discharge data, including diagnosis (coded with the International Classification of Diseases 10th revision, ICD-10). Dispensed drugs are encoded using the Anatomic, Therapeutic, Chemical (ATC) classification [17].

### 2.3. Study Population

The study period for the cross-sectional analysis was the year 2016. Individuals from the EGB were considered eligible for the analysis providing that they were covered by French healthcare insurance from 1 January 2015, and that they were alive and aged 45 or older on 1 January 2016. Individuals who died in 2016 were not excluded in order to take into account end of life polypharmacy. The included individuals were classified according to age on 1 January 2016, as older adults (65 and over at that date) or middle-aged adults (45 to 64 years old at that date).

### 2.4. Drugs of Interest

For our study of chronic polypharmacy, all drugs but those belonging to the following classes were considered: homeopathy, vaccines (ATC: J07), anesthetics (N01), blood substitutes and perfusion solutions (B05), immune sera and immunoglobulins (J06), diagnostic drugs or radiopharmaceuticals (S01J, V04, and V09), contrast media (V08), antidotes (V03AB), medical gases (V03AN), insecticides and repellents (P03B), antiseptics and disinfectants (D08), medicated dressings (D09), surgical dressings (V20), tissue adhesives (V03AK), and all other non-therapeutic products (V07). Drugs were identified using their ATC codes; information on fixed-dose combinations was split to allow taking into account, independently, the use of each of the combined active ingredients.

### 2.5. Exposure

Chronic drug use was defined as uninterrupted daily use lasting at least 183 days. Chronic polypharmacy was defined as the chronic use of five drugs or more, and chronic hyperpolypharmacy as the chronic use of ten drugs or more. The identification of chronic drug use and polypharmacy for each subject was based upon information regarding the drug prescriptions they had during the year 2016 and 2015.

For each identified prescription, treatment use was assumed to start on the date of prescription. From this date, the length of treatment episodes was computed using estimates of the mean durations of drug prescriptions in France [18]. A 20% grace period was added to each theoretical duration of prescription to take into account potential compliance issues [19].

As information on drugs prescribed during hospitalization was not available in EGB; these were assumed to be similar to those used on the day before admission [20].

### 2.6. Potentially Inappropriate Medications

PIM lists, originally aimed at helping clinicians, can also be used for research purposes [21,22,23,24]. These lists were created by panels of experts and are key indicators of medication prescribing quality. They list specific drugs that should not be prescribed for general treatment or in patients with several conditions.

In this study, PIM were identified using two tools specifically developed to assess PIM in older adults (≥65 years old): the Beers criteria list and its adaptation to the French setting, the Laroche list, and one developed for the assessment of PIM in middle-aged adults (45 to 64 years old), the PROMPT (Prescribing Optimally in Middle-aged People’s Treatments) criteria list [25,26,27]. For individuals who turned 65 in 2016, the Beers and Laroche lists were used throughout the 2016 period.

As the EGB database included limited medical information and does not include information on lab results, not all criteria listed in those tools could be used and some were only partially applicable. The used criteria, their requirements, their applicability, and the ATC codes considered for the identification of the corresponding drugs are listed in the Appendix A (Appendix A presents the combined Beers and Laroche criteria; Appendix A presents the PROMPT criteria; Appendix A presents the overlaps of Appendix A). Two levels of PIM were thus defined: PIM-narrow, considering only fully applicable criteria, and PIM-broad, considering both fully and partially applicable criteria.

### 2.7. Descriptive Analyses

The prevalence of chronic polypharmacy was defined as the proportion of individuals who had at least one day of chronic polypharmacy in 2016; the prevalence of chronic hyperpolypharmacy was defined similarly. In individuals with chronic polypharmacy/hyperpolypharmacy, the prevalence of chronic PIM was defined as the proportion of individuals with at least one day with any chronic PIM use in 2016. Estimations were carried out considering the two defined levels of PIMs—PIM-narrow and PIM-broad. The most frequent PIMs were defined as chronic PIMs contributing to polypharmacy/hyperpolypharmacy in at least 1% of individuals with such. Cumulated exposure to chronic PIM within polypharmacy/hyperpolypharmacy was defined as the proportion of days of exposure to PIM over the number of days of exposure to all drugs contributing to any individual’s period of chronic polypharmacy/hyperpolypharmacy for the year 2016.

All the analyses were stratified according to chronic polypharmacy (polypharmacy vs. hyperpolypharmacy) and according to age (older adults vs. middle-aged).

All the data management and analyses were performed using SAS software version 9.4 (SAS Institute, Cary, NC, USA).

Neither committee approval nor informed consent was required because only anonymous data were used under the French Data Protection Supervisory Authority (CNIL) agreement.

## 3. Results

### 3.1. Description of the Study Population

Within the EGB, we identified 276,788 individuals who met the eligibility criteria. The sex ratio was close to 1; the mortality in 2016 was 1.5% (Table 1). A total of 117,545 were older adults (≥65 years old) and 159,243 were middle-aged adults (45–65 years old). In 2016, the prevalence of chronic polypharmacy and hyperpolypharmacy increased with age, from 5.4% and 0.5% in middle-aged adults to 23.7% and 2.4% in older adults. Diabetes mellitus was the most frequent long-term disease in both older adults (14.5%) and middle-aged adults (6.0%) (Table 1, Appendix A).

### 3.2. Potentially Inappropriate Medications in Chronic Polypharmacy

In 2016, among older adults with chronic polypharmacy 18,036 (64.8%) had at least one chronic PIM, as defined by the Beers/Laroche criteria. Among older adults with chronic hyperpolypharmacy, 2544 (88.7%) had at least one chronic PIM. Overall, chronic PIMs represented 13.5% of the total exposure to drugs involved in chronic polypharmacy. Pump proton inhibitors (PPIs) used without any concomitant use of chronic nonsteroidal anti-inflammatory drugs (NSAIDs) or corticosteroids were the most frequent chronic PIMs (43.4% of older adults with chronic polypharmacy; 67.1% of older adults with chronic hyperpolypharmacy; 6.3% of total exposure to chronic drugs). These were followed by short- and intermediate-acting benzodiazepines (13.7% of older adults with chronic polypharmacy; 23.0% of older adults with chronic hyperpolypharmacy; 2.0% of total exposure) and hypnotics (6.1% of older adults with chronic polypharmacy; 13.3% of older adults with chronic hyperpolypharmacy; 0.8% of total exposure) (Table 2. Full results are available in Appendix A, descriptions of criteria are presented in Appendix A).

In 2016, among middle-aged adults with chronic polypharmacy 4009 (46.2%) had at least one chronic PIM, as defined by the PROMPT criteria. Among middle-aged adults with chronic hyperpolypharmacy, 570 (75.0%) had at least one chronic PIM. Overall, chronic PIMs represented 10.4% of the total exposure to drugs involved in chronic polypharmacy. Short- and intermediate-acting benzodiazepines were the most frequent PIMs (16.1% of middle-aged adults with chronic polypharmacy; 30.5% of adults with chronic hyperpolypharmacy; 2.7% of total exposure to chronic drugs). They were followed by long-acting sulfonylureas (12.3% of middle-aged adults with chronic polypharmacy; 23.4% of middle-aged adults with chronic hyperpolypharmacy; 1.9% of total exposure) and long-acting benzodiazepines (10.1% of middle-aged adults with chronic polypharmacy; 18.2% of middle-aged adults with chronic hyperpolypharmacy; 1.5% of total exposure) (Table 3. Full results are available in Appendix A, description of criteria are in Appendix A).

## 4. Discussion

In this study, we found that the prevalence of PIM in older adults, defined according to the Beers and Laroche criteria, was substantial and increased with the number of medications involved in chronic polypharmacy (64.8% of older adults with chronic polypharmacy and 88.7% with chronic hyperpolypharmacy). We also observed this trend with the PROMPT criteria in middle-aged individuals (46.2% of middle-aged individuals with chronic polypharmacy and 75.0% with chronic hyperpolypharmacy). The most frequent PIM were PPIs, benzodiazepines and derivatives, long-acting sulfonylureas, opioids, central alpha-agonists, and antidepressants.

In the literature, an increase in the prevalence of PIM with polypharmacy has already been observed in both inpatients and ambulatory patients [28,29], especially in the elderly [30,31]. Similarly, PPIs, benzodiazepines, and sulfonylureas are frequent PIMs reported.

PPIs were the most frequent PIM in a recent study based on the Beers criteria, which found that gastrointestinal medications—referring to metoclopramide, mineral oil, or PPIs—were the most frequent PIMs (35.6% of adults) [28]. In Ireland, PPIs above maintenance dosage for greater than 8 weeks were the second most frequent PIM, according to the PROMPT criteria [32]. We could not assess inappropriate PPIs in middle-aged adults because the PROMPT criteria required the maintenance dose, and this information was not available from the EGB. However, a previous study showed that almost a quarter (16 million people) of the French population had a prescription of PPIs in 2015. Among them, half started the treatment to prevent adverse gastrointestinal events. However, 80% did not need this type of prevention according to the guidelines [33]. In 2012, PPIs were among the highest-selling classes of drugs in the United States, and esomeprazole was the top selling of all drugs [34]. Our study showed that PPIs are often prescribed chronically and inappropriately to patients who are already using many chronic treatments. PPIs can be associated with various adverse drug reactions such as fractures or infections, even if these occurrences are rare [35,36]. Altogether, these results highlight the need for a more rational use of PPIs. Targeting PPIs in polypharmacy could be an efficient first step to reduce their use, as these are the most prevalent among patients.

Benzodiazepines and derivatives are also psychotropic drugs often reported to be overwhelmingly used. France is, for instance, the country with the second highest level of benzodiazepine use in Europe, with an overall prevalence of long-term use estimated at 15% in new users [37,38]. This number is preoccupying especially because it could involve patients with multiple comorbidities and increase the risk of adverse drug reactions [39,40]. Although these drugs are frequently prescribed at the end of life, the low all-cause mortality in 2016 and our definition of chronic drug use highlighted the fact that benzodiazepines mainly concern healthy people.

Long-acting sulfonylureas also constitute a major concern. A population-based study in the United Kingdom showed that sulfonylureas, even as a second-line treatment, were associated with an increased risk of myocardial infarction, all-cause mortality, and severe hypoglycemia, compared with metformin monotherapy [41]. In older adults, the Beers criteria defined anticholinergic antidepressants, such as tricyclic antidepressants or paroxetin, as also of concern, especially because other selective serotonin reuptake inhibitors (SSRIs) or classes of antidepressants might be appropriate [25]. These two classes (i.e., long-acting sulfonylureas and anticholinergic antidepressants) were examples of classes for which interventions aiming to reduce PIM should lead to treatment switches, which would result in improving the appropriateness of patients’ treatment without lowering the importance of chronic polypharmacy.

Our study used the Beers, Laroche, and PROMPT criteria, but researchers can use many other tools to identify PIMs. A metanalysis on this topic has referenced 36 different PIM lists, and many were derived from the Beers criteria [24]. The Beers and Laroche criteria defined drugs that are potentially inappropriate but not definitely inappropriate. This means that these drugs should be avoided in most individuals, according to the recommendations and statements listed, but some exceptions could exist [42]. Although PIMs could be appropriate in particular and rare situations, the high prevalence of the main PIMs found in our study seemed incompatible with only appropriate use in clinical conditions.

This study was completed with some strengths and limitations that are mainly related to the nature of the database used. The EGB database provided an exhaustive recording of outpatient medical reimbursements, hospitalization diagnosis codes, and lengths of stay, which allowed us to consider all of the drug reimbursements of each subject in 2016, enabling us estimate chronic polypharmacy and potentially inappropriate medication. In addition, the EGB database is highly representative of the French population [17]. Conversely, the durations of prescriptions were not available from the EGB. However, this information existed owing to a repeated dedicated survey focusing on prescription lengths performed from a representative sample of French physicians, the results of which were used to estimate treatment durations in our study [18]. The EGB did not include over-the-counter (OTC) drugs, as they are not reimbursed, which was a limitation in the complete identification of drug treatments. If this was of theoretical importance, this might have had a negligible impact in our study that focused on chronically used drugs—i.e., drugs used continuously for at least six months. Finally, the most important limitation was that the existing PIM lists did not allow for the identification of all inappropriate situations. Indeed, a study evaluating the performance of the Beers criteria showed that clinicians and pharmacists would prescribe inappropriate drugs twice more than the Beers criteria recommended [43]. Consequently, we combined the Beers list with the Laroche list, which reflects the opinion of different experts with a strong knowledge of French clinical practice, and used a pharmacological evaluation for the selection of PIM [26]. However, using data from the EGB, we could only partially apply the PIM criteria because not all the required information (clinical, lab, maintenance dosage) was available from the EGB. In this aspect, our estimate of the prevalence of chronic PIM within individuals with polypharmacy is likely to constitute an underestimation. The awareness of deprescribing has been rising recently. However, the major PIMs we identified in 2016 will continue to be a major concern in 2020 since their use and prescription may also be due to high patient demands. Longitudinal studies quantifying PIMs over the last few years are required. This study was reported using the observational routinely collected health data statement for pharmacoepidemiology (RECORD-PE; Appendix A) [44].

## 5. Conclusions

Our results provided us with two important types of information. First, chronic polypharmacy, and chronic PIM within it, was highlighted as an important concern already in middle-aged adults, showing that interventions aiming to reduce the burden of PIM should not be limited to older adults. Second, it helped in identify the drugs most involved in chronic PIM use both in middle-aged and older adults and that should be primarily targeted by such interventions. From this perspective, PPIs and benzodiazepine appeared once again as a major feature in drug overuse, extending to patients with an already high drug intake burden.

## Figures and Tables

**Table 1 jcm-09-03728-t001:** Description of demographic characteristics, chronic polypharmacy, and chronic diseases at baseline according to age.

	Cohort *n* = 276,788	Middle-Aged ^1^ *n* = 159,243	Older Adults ^2^ *n* = 117,545
Age, mean (sd)	63.5 (10.2)	54.1 (5.7)	76.2 (14.2)
Sex, *n* (%)			
Men	131,275 (47.4)	79,920 (50.2)	51,355 (43.7)
Women	145,513 (52.6)	79,323 (49.8)	66,190 (56.3)
Dead in 2016, *n* (%)	4239 (1.5)	663 (0.4)	3576 (3.0)
Chronic polypharmacy *n* (% ± 95CI)	36,500 (13.2 ± 0.2)	8666 (5.4 ± 0.1)	27,834 (23.7 ± 0.2)
Chronic hyperpolypharmacy *n* (% ± 95CI)	3628 (1.3 ± 0.0)	760 (0.5 ± 0.0)	2868 (2.4 ± 0.0)
Most frequent chronic diseases ^3^			
Diabetes (type 1 or type 2)	26,622 (9.6)	9602 (6.0)	17,020 (14.5)
Cancer or leukemia	21,991 (8.0)	6914 (4.3)	15,077 (12.8)
Coronary artery disease	13,248 (4.8)	3566 (2.2)	9682 (8.2)
Heart failure, arrhythmia or valvular heart disease	12,437 (4.5)	1908 (1.2)	10,529 (9.0)
Psychiatric diseases	10,108 (3.7)	6301 (4.0)	3807 (3.2)

^1^ 45–65 years old; ^2^ ≥65 years old; ^3^ defined as presenting with a prevalence >3% in the population; sd: standart deviation; 95CI: 95% confidence interval.

**Table 2 jcm-09-03728-t002:** Prevalence and exposure to the most frequent potentially inappropriate medications in older adults with chronic polypharmacy according to the Beers criteria and Laroche list.

	Older Adults with Chronic Polypharmacy	Older Adults with Chronic Hyperpolypharmacy	Cumulated Exposure to PIMs in Older Adults with Chronic Polypharmacy (%)
	*n* = 27,834	*n* = 2868	
Potentially inappropriate medications-broad ^1^	18,036 (64.8)	2544 (88.7)	13.5
Potentially inappropriate medications-narrow ^2^	10,220 (36.7)	1730 (60.3)	6.7
Proton pump inhibitors (PPIs) without chronic use of nonsteroidal anti-inflammatory drugs (NSAIDs) or corticosteroids ^1^	12,073 (43.4)	1924 (67.1)	6.3
Benzodiazepines—short- and intermediate-acting	3807 (13.7)	660 (23.0)	2.0
Hypnotics (z-drugs)	1688 (6.1)	382 (13.3)	0.8
Central alpha-agonists	1404 (5.0)	308 (10.7)	0.8
Antidepressants (Tricyclic antidepressants (TCAs)/Paroxetine)	1324 (4.8)	228 (8.0)	0.7
Benzodiazepines—long-acting	1271 (4.6)	286 (10.0)	0.6
Sulfonylureas—long-acting	1071 (3.9)	201 (7.0)	0.6
First-generation antihistamines	659 (2.4)	159 (5.5)	0.5
Anticholinergic antispasmodics	626 (2.3)	119 (4.2)	0.3
Antidepressants (TCA, Selective serotonin reuptake inhibitors (SSRIs), or Serotonin–norepinephrine reuptake inhibitors (SNRIs)) with history of falls or fractures ^1^	420 (1.5)	73 (2.6)	0.2
Non-cyclooxygenase-selective NSAIDs, oral without PPI ^1^	418 (1.5)	65 (2.3)	0.2
Ergoloid mesylates	380 (1.4)	71 (2.5)	0.2

PPIs: proton pump inhibitors; NSAIDs: nonsteroidal anti-inflammatory drugs; TCAs: Tricyclic antidepressants; SSRIs: Selective serotonin reuptake inhibitors; SNRIs: Serotonin–norepinephrine reuptake inhibitors. Data are expressed as n (%); ^1^ considering both fully and partially applicable criteria; ^2^ considering only fully applicable criteria; ^1,2^ more details are provided in Appendix A.

**Table 3 jcm-09-03728-t003:** Prevalence and exposure to the most frequent potentially inappropriate medications in middle-aged adults with chronic polypharmacy according to the PROMPT criteria.

	Middle-Aged Adults with Chronic Polypharmacy	Middle-Aged Adults with Chronic Hyperpolypharmacy	Cumulated Exposure to PIM in Middle-Aged Adults with Chronic Polypharmacy (%)
	*n* = 8666	*n* = 760	
Potentially inappropriate medications	4009 (46.2)	570 (75.0)	10.4
Benzodiazepines—short- and intermediate-acting	1395 (16.1)	232 (30.5)	2.7
Sulfonylureas—long- acting	1069 (12.3)	178 (23.4)	1.9
Benzodiazepines—long- acting	879 (10.1)	138 (18.2)	1.5
Opioid (use without laxative)	639 (7.4)	143 (18.8)	1.1
Hypnotics (z-drugs)	637 (7.4)	115 (15.1)	1.0
First generation antihistamines	450 (5.2)	90 (11.8)	0.7
Association of esomeprazole/omeprazole and clopidogrel ^1^	251 (2.9)	59 (7.8)	0.8
Oral corticoid (without use of bisphosphonate)	176 (2.0)	38 (5.0)	0.3
Tricyclic antidepressants in first-line treatment	107 (1.2)	16 (2.1)	0.2
Chronic NSAIDs	80 (0.9)	14 (1.8)	0.1

NSAIDs: nonsteroidal anti-inflammatory drug; PIM: potentially inappropriate medications; PROMPT: Prescribing Optimally in Middle-aged People’s Treatments; Data are expressed as n (%). ^1^ These criteria considered both drugs (esomeprazole or omeprazole and clopidogrel) as potentially inappropriate, so the density (0.8%) of exposure is twice the exposure of each drug individually (0.4% each).

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
