# Peer review of "The Burden of Potentially Inappropriate Medications in Chronic Polypharmacy"

_jcm, 2020, doi:10.3390/jcm9113728_

Round 1
Reviewer 1 Report
The article is well written, has a logic order, methods and results are clear, data are well presented, and the discussion well builds on the information before. I have no comments.
from a purely scientific / methodological approach I think it is an excellent article. It reads easy without any need for clarifications. Abstract, introduction, methods contain the expected information. Graphics are logic and well presented, conclusion flows well from the results etc.
Author Response
We would like to warmly thank the reviewer#1 for his positive evaluation
Reviewer 2 Report
This was cross-sectional study of a sample of middle aged and older adults from a national database in France in 2016 that contains prescription information. They found both categories of age groups had significant chronic use of PIMs such as PPIs, benzodiazepines, and long-acting sulfonylureas.
Major critiques:
- Methods, Study population: By including individuals at end of life, how do you think that influenced your results? Did you look at an sub-analysis of patients who died to see if their medication use patterns differed? We often use benzos and opioids at end of life for symptom relief.
- Results, Table 1: In addition to showing the % of chronic polypharmacy and hyperpolypharmacy, it would be informative to include the median and interquartile range (or mean/std deviation if more appropriate) of the number of medications for middle aged and elders.
- Discussion: Since these data were from 2016, do you think prescribing in France has changed in the last several years? There has been much more awareness of deprescribing in recent years. Have their been national initiatives in France to reduce polypharmacy?
Minor critiques:
- I applaud the authors for their excellent command of English; however, there are several word choice and verb tenses that will need copyediting.
- Rather than the word elders, I believe geriatricians prefer "older adults".
- Discussion, 6th paragraph, last sentence: It is unclear what he authors are saying in this sentence--"Even if sometimes the medication could be appropriate, the high prevalence of some PIMs found in our study seem not compatible with the only use in appropriate clinical conditions." Does this mean that most medications were used inappropriately?
Author Response
Reviewer #2, Major critics #1
Methods, Study population: By including individuals at end of life, how do you think that influenced your results? Did you look at an sub-analysis of patients who died to see if their medication use patterns differed? We often use benzos and opioids at end of life for symptom relief.
Authors:
Thank you for this valuable comment. Although this is not among the objectives of this study, it would be very interesting to perform an analysis at end of life. However, as we focused on chronic drug use, defined by a continuous use of 6 months, we think that an increase of chronic benzodiazepines or opioids uses before death would not have had a major impact on presented prevalences of PIMs. Moreover, all-cause mortality in 2016 was only of 1.5% in the studied population. We discussed this point (lines 200 - 202).
Reviewer #2, Major critics #2
Results, Table 1: In addition to showing the % of chronic polypharmacy and hyperpolypharmacy, it would be informative to include the median and interquartile range (or mean/std deviation if more appropriate) of the number of medications for middle aged and elders.
Authors: Unfortunately, we do not have access to the data anymore and these analyses can not be performed now. However, if the editor thinks that it is essential for publication we can set up new access, but the submission deadline would have to be largely extended.
Reviewer #2, Major critics #3
Discussion: Since these data were from 2016, do you think prescribing in France has changed in the last several years? There has been much more awareness of deprescribing in recent years. Have their been national initiatives in France to reduce polypharmacy?
Authors: This is a very interesting question. Indeed, clinicians are probably more sensitive to deprescription. However the major PIMs we identified have been a problem for years. PPIs and benzodiazepines uses are also mainly due to patients’ demands. To answer this question we would need to conduct a longitudinal study describing the changes of prevalence of polypharmacy and PIMs over the last years. We added this point in the discussion (lines 241 - 244)
Reviewer #2, Minor critics #1
I applaud the authors for their excellent command of English; however, there are several word choice and verb tenses that will need copyediting.
Authors: Thank you for this comment. We made corrections
Reviewer #2, Minor critics #2
Rather than the word elders, I believe geriatricians prefer "older adults".
Authors: Thank you. We changed it
Reviewer #2, Minor critics #3
Discussion, 6th paragraph, last sentence: It is unclear what the authors are saying in this sentence--"Even if sometimes the medication could be appropriate, the high prevalence of some PIMs found in our study seem not compatible with the only use in appropriate clinical conditions." Does this mean that most medications were used inappropriately?
Authors: We understand the question of Reviewer#2. Sometimes medications defined as PIMs by explicit criteria are actually appropriate in particular and rare situations. This is why they are defined as potentially inappropriate. However the high prevalences we observed cannot be justified by these particular situations. We clarified this point in the discussion section (lines 217-219).
Reviewer 3 Report
Thank you for providing the opportunity to review this manuscript. The manuscript is well written and add to the literature on the need to reduce PIMs.
Below are some comments from the review
- As the lists that are used to assess PIMs are explicit in nature. Suggest a table that highlights the overlap in the lists for older people >65 years vs middle aged individuals
- Suggest using RECORD-PE checklist and reporting
Author Response
Thank you for this comment. To address these points, we added two tables in the supplementary materials (Table S3 and S8). Mention to RECORD PE was added in the discussion section (lines 244 - 246)